# Anti-Inflammatory and Antibacterial Effects and Mode of Action of Greek Arbutus, Chestnut, and Fir Honey in Mouse Models of Inflammation and Sepsis

**DOI:** 10.3390/microorganisms10122374

**Published:** 2022-11-30

**Authors:** Elisavet Stavropoulou, Eleftheria Ieronymaki, Evangelia Dimitroulia, Theodoros C. Constantinidis, Georgia Vrioni, Christos Tsatsanis, Athanasios Tsakris

**Affiliations:** 1Department of Microbiology, Medical School, National and Kapodistrian University of Athens, 11527 Athens, Greece; 2Service of Infectious Diseases, Department of Medicine, Lausanne University Hospital and University of Lausanne (CHUV), 1011 Lausanne, Switzerland; 3Laboratory of Clinical Chemistry, Medical School, University of Crete, 71003 Heraklion, Greece; 4Institute for Molecular Biology and Biotechnology, Foundation for Research and Technology (IMBB-FORTH), 71500 Heraklion, Greece; 5Laboratory of Hygiene and Environmental Protection, Department of Medicine, Democritus University of Thrace, 68100 Dragana Alexandroupolis, Greece

**Keywords:** sepsis, honey, inflammation, bactericidal, antimicrobial, mice, LPS, cecal slurry-induced peritonitis, TNF-a, iNOS, natural compounds, Manuka

## Abstract

**Background:** Honey has been shown to possess anti-inflammatory and bactericidal properties that may be useful for the prevention and treatment of infections as well as of acute and chronic inflammatory diseases. The antimicrobial potency of honey could be attributed to its physicochemical characteristics combined with the presence of certain compounds, such as hydrogen peroxide and polyphenols. Honey’s bacteriostatic or bactericidal capacity varies depending on its composition and the bacterial type of each infection. Nevertheless, not all honey samples possess anti-inflammatory or antibacterial properties and their mechanism of action has not been clearly elucidated. **Objectives:** We therefore investigated the anti-inflammatory properties of three different honey samples that derived from different geographical areas of Greece and different botanical origins, namely, arbutus, chestnut, and fir; they were compared to manuka honey, previously known for its anti-inflammatory and antibacterial activity. **Materials and Methods:** To test the anti-inflammatory activity of the different samples, we utilized the in vivo model of LPS-driven inflammation, which induces septic shock without the presence of pathogens. To evaluate the antibacterial action of the same honey preparations, we utilized the cecal-slurry-induced peritonitis model in mice. Since acute inflammation and sepsis reduce the biotransformation capacity of the liver, the expression of key enzymes in the process was also measured. **Results:** The administration of all Greek honey samples to LPS-stimulated mice revealed a potent anti-inflammatory activity by suppressing the TNFα serum levels and the expression of TNFα and iNOS in the liver at levels comparable to those of the manuka honey, but they had no effect on IL-6 or IL-1β. It was shown that the LPS-induced suppression of CYP1A1 in the liver was reversed by Epirus and Crete fir honey, while, correspondingly, the suppression of CYP2B10 in the liver was reversed by Evros chestnut and Epirus fir honey. The effect of the same honey samples in polymicrobial peritonitis in mice was also evaluated. Even though no effect was observed on the disease severity or peritoneal bacterial load, the bacterial load in the liver was reduced in mice treated with Evros chestnut, Epiros fir, and Crete fir, while the bacterial load in the lungs was reduced in Epirus arbutus, Crete fir, and manuka honey-treated mice. **Conclusion:** Our findings suggest that these specific Greek honey samples possess distinct anti-inflammatory and antibacterial properties, as evidenced by the reduced production of pro-inflammatory mediators and the impaired translocation of bacteria to tissues in septic mice. Their mode of action was comparable or more potent to those of manuka honey.

## 1. Introduction

Inflammation is a protective response of the immune system that can be triggered by pathogenic and nonpathogenic stimuli, such as bacteria, fungi, parasites, toxic substances, and tissue injury. Nevertheless, the onset of uncontrolled inflammation along with the presence of oxidative stress could lead to the development of chronic inflammatory conditions [1]. A bacterial infection could lead to systemic inflammation and sepsis. Sepsis is a significant clinical challenge in intensive care that frequently leads to multiorgan failure and high mortality rates [2]. Although there are well established empirical or targeted treatment options against the underlying infection antibiotic resistance emerges, and the development of new therapeutic strategies, identification of potential drug candidates and antibiotic alternatives is essential to prevent and combat sepsis, resolve inflammation and maintain homeostasis.

Honey is a nutritional healthy product with known immunomodulatory, antioxidant and antimicrobial activity that can be used as an alternative treatment in a wide range of human acute or chronic inflammatory disorders, including wound healing, asthma, type 2 diabetes, cardiovascular and gastrointestinal diseases [3]. Composition of honey is important reflecting its anti-inflammatory and antimicrobial properties. It consists of macronutrients and micronutrients, including sugar, minerals, vitamins, protein, enzymes and polyphenols, which vary according to the botanical source, honey bee species, environmental conditions and geographical origins [4,5,6].

Anti-inflammatory properties of honey could be attributed to its phenolic and flavonoid compounds, while other sugar and nonsugar ingredients are known to exert immunomodulatory effects [7]. A plethora of in vivo and in vitro studies delineate the potential anti-inflammatory mechanisms of honey and its constituents that involve downregulation of NF-κB and MAPK signaling pathways leading to decreased levels of pro-inflammatory cytokines, including tumor necrosis factor α (TNF-α), interleukin-6 (IL-6) and IL-1β [8,9]. The expression of pro-inflammatory enzymes such as COX, implicated in metabolism of arachidonic acid and production of prostaglandins (PG), and iNOS, catalyzing for the production of nitric oxide (NO), were also suppressed [8].

A broad spectrum of Gram-negative and Gram-positive antibiotic-sensitive and resistant bacteria were found to be sensitive to honey. Antibacterial effectiveness and potency of honey could be related to its physicochemical characteristics [10]. Low water activity and high sugar concentration could cause osmotic stress to microorganisms and inhibit their proliferation and growth [10,11]. Acidity and low pH due to the presence of organic acids increases its antimicrobial efficacy [10,12]. In addition, certain compounds of honey including hydrogen peroxide, phenolic acids and flavonoids, methylglyoxal and bee defensin-1 were found to be potent antimicrobial agents [13,14]. The mode of action of honey depends on the type of bacteria Gram-positive or Gram-negative bacteria and its composition. Honey could cause morphological and structural alterations on shape, size and flagella of bacteria [15,16], inhibition of cell cycle [15], membrane depolarization [17], metabolic disruption [18], prevention of biofilm formation [19,20], blockage of efflux pumps activity [17], reduction of virulence and quorum sensing [21,22] and reduced response of bacteria to stress [23].

Manuka honey gained much attention since it possesses a unique antimicrobial efficacy and potency that was found to be related mainly to its methylglyoxal content [24]. Studies demonstrated its ability to cover a wide range of bacteria and fungi, including *Staphylococcus aureus*, *Escherichia coli*, *Klebsiella pneumoniae*, *Pseudomonas aeruginosa*, *Enterobacter cloacae* and their antibiotic resistant counterparts, although Gram-positive bacteria were found to be more sensitive to Manuka honey compared to Gram-negative bacteria [24]. Due to the fact of its potent activity, Manuka honey has been approved as medical-grade honey used for the treatment of surface wounds and burns. Nevertheless, several studies on the antibacterial efficacy of other honey varieties showed advantages over or similarities with Manuka honey [25].

Previous evidence showed that Greek honey types derived from different botanical origins and geographical locations displayed antibacterial properties in vitro against Gram-negative multidrug-resistant bacteria, such as *Enterobacter cloacae*, *Klebsiella pneumoniae*, and *Pseudomonas aeruginosa* [26]. The aim of this study was to further evaluate the in vivo anti-inflammatory and antibacterial potential of these honey varieties, namely, arbutus honey, chestnut honey, and two fir honey samples, while Manuka honey served as the control, using mouse models of acute inflammation and polymicrobial peritonitis.

## 2. Materials and Methods

Eight-week-old female C57BL/6 mice were kept in the pathogen-free animal facility of the School of Medicine, University of Crete, Heraklion, Crete. All procedures were conducted in compliance with the protocols approved by the Animal Care Committee of the University of Crete, School of Medicine (Heraklion, Crete, Greece), and from the Veterinary Department of the Region of Crete (Heraklion, Crete, Greece).

### 2.1. LPS-Induced Endotoxin Shock

A minimal lethal dose of LPS (1.5 mg/25 g of body weight) was used as described previously [27,28]. The injected animals were monitored for 4 h, and they were sacrificed when they were moribund. The mice were pretreated with normal saline or a 30% honey concentration (i.p. injection) 30 min before the LPS injection. Serum was collected at 1 and 4 h after the LPS stimulation and stored at −80 °C until ELISAs were performed.

### 2.2. Cecal-Slurry-Induced Polymicrobial Sepsis

Polymicrobial sepsis was induced using the fecal peritonitis method as previously described [29,30]. Briefly, the cecum contents of adult C57BL/6 mice (6–8 weeks old) were suspended in a 5% dextrose solution at a final concentration of 80 mg/mL and then passed through a 70 nm filter. The aliquots were stored at −80 °C until challenge. An amount of 1.1 mg/per gram of mouse bodyweight of the CS was injected intraperitoneally (i.p.) The age-matched mice received an equal amount of 5% dextrose and served as the controls. The injected animals were monitored for 24 h, and then they were sacrificed. The mice were pretreated with normal saline or a 30% honey concentration (i.p. injection) 30 min before the CS injection. Serum was collected at 4 and 24 h after the cecal slurry challenge. A peritoneal lavage was also collected. All samples were stored at −80 °C until ELISAs were performed.

### 2.3. ELISA

The cytokine concentration for TNFα and IL-6 was determined by ELISA at the indicated time points using ELISA kits (R&D Systems) according to the manufacturer’s instructions.

### 2.4. RNA Isolation and Quantitative PCR

The RNA from the liver was isolated using TRIzol reagent (Life Technologies, Carlsbad, CA, USA). One microgram of total RNA was used for the cDNA synthesis (TAKARA, Shiga, Japan). The SYBR Green method was followed in the PCR reaction. Actin served as the housekeeping gene. The annealing was carried out at 60 °C for 30 s and denaturation at 95 °C for 15 s for 40 cycles in a 7500 Fast Real-Time PCR System (Life Technologies/Applied Biosystems, Carlsbad, CA, USA). The amplification efficiencies were the same as the one for Actin, as indicated by the standard curves of the amplification, which allowed us to use the following formula: fold difference = 2^−ΔCtA–ΔCtB^, where Ct is the cycle threshold.

### 2.5. Statistical Analysis

The samples collected from mice were analyzed in parallel to determine the respective biomarkers or parameters. The data were analyzed using one-way ANOVA with the Bonferroni multiple comparison post-test, using GraphPad InStat software (GraphPad, San Diego, CA, USA). Data that included multiple time points for each parameter (i.e., sepsis score and sepsis severity) were analyzed using two-way ANOVA. The results are expressed as the mean ± SEM or as the median (minimum to maximum) as indicated. The *p*-values < 0.05 were considered significant.


**Primers for RT-PCR**

TNF-α

Fwd
5′-CACGCTCTTCTGTCTACTGAACTTCG-3′Rev5′-GGCTGGGTAGAGAATGGATGAACACC-3′IL-6Fwd5′-TTCCATCCAGTTGCCTTCTT-3′Rev5′-CAGAATTGCCATTGCACAAC-3′IL-1βFwd5′-GGATGAGGACATGAGCACCT-3′Rev5′-TCCATTGAGGTGGAGAGCTT-3′iNOSFwd5′-TCCTGGAGGAAGTGGGCCGAAG-3′Rev5′-CCTCCACGGGCCCGGTACTC-3′CYP1A1Fwd5′-TTAACCATGACCGGGAACTGT-3′Rev5′-CACTTTCGCTTGCCCAAACC-3′CYP2B10Fwd5′-GCTCATTCTCCAGCCAGATGTT-3′Rev5′-CTCCATGCGCAGAAGGTAAA-3′CYP3A11Fwd5′-TTCACCCTGCATTCCTTGGC-3′Rev5′-TACGTGGGAGGTGCCTTGTT-3′CYP3A25Fwd5′-ATCTTCGGGGGCTATGATGC-3′Rev5′-AGGTGACAGGTGCCTTATTGG-3′ActinFwd5′-CATTGCTGACAGGATGCAGAAGG-3′Rev5′-TGCTGGAAGGTGGACAGTGAGG-3′

## 3. Results

### 3.1. Effect of Greek Honey on Pro-Inflammatory Cytokines in the LPS-Induced Inflammation and Sepsis Model In Vivo

To investigate the anti-inflammatory potential of the Greek honey samples, an in vivo model of lipopolysaccharide (LPS)-induced acute aseptic inflammation in mice was used. Increased levels of pro-inflammatory cytokines IL-6 and TNF-α were observed in mice treated with LPS 1 h and 4 h after injection compared to the control mice (Figure 1A–F). The LPS-challenged mice pretreated with all the different types of honey showed significantly lower levels of serum TNF-α compared to mice treated with LPS in both time points tested (Figure 1A–C).

In accordance, the TNFα mRNA levels significantly decreased in the liver of the mice treated with honey and LPS (Figure 1G). The LPS increased the expression of iNOS, an important pro-inflammatory mediator. iNOS expression was significantly reduced in the liver of LPS-stimulated mice pretreated with all different honey types compared to the LPS-treated group (Figure 1H). The liver expression and serum cytokine levels of the IL-6 levels and IL-1β did not show any difference among all LPS-stimulated groups (Figure 1D–F,I,J).

The evaluation of the clinical status of mice showed that mice receiving LPS displayed an impaired health status compared to the controls, as evidenced by the increased clinical severity score (Figure 2A). The treatment with different types of honey did not show any effect on the disease severity (Figure 2A).

Sepsis is associated with disturbances in glucose regulation and the occurrence of hypoglycemia. Therefore, disease progression was also evaluated by measuring the blood glucose concentration. The levels of glucose in the serum significantly declined after 4 h compared to 1 h post-LPS injection (Figure 2B–D). No significant difference was observed among all LPS-injected groups (Figure 2C).

Systemic inflammation could influence multiple organ functions, including the liver. Liver function was evaluated by measuring its biotransformation capacity. CYPs are important enzymes expressed in the liver, and they are responsible for the oxidation and detoxification of various compounds. Endotoxin induced a significant reduction in the expression of CYPs in the liver of LPS-stimulated mice compared to the controls (Figure 2E–H). The CYP1A1 mRNA levels significantly increased in the liver of the mice that received honey derived from Epirus fir and Crete fir before an LPS injection compared to the LPS-treated group (Figure 2E). In addition, the expression of CYP2B10 was found to be elevated in the liver of mice treated with honey derived from Evros chestnut and Epirus fir, compared to the control LPS-stimulated group (Figure 2F). The expression of CYP3A11 and CYP3A25 was not significantly altered in all honey pretreated groups (Figure 2G,H).

### 3.2. Evaluation of the Antibacterial Properties of Greek Honey in an In Vivo Model of Polymicrobial Sepsis in Mice

In light of the anti-inflammatory action of Greek honey in vivo, we further investigated the antibacterial properties of the same honey samples using an in vivo model of cecal-slurry (CS)-induced polymicrobial peritonitis in mice. Sepsis, as a consequence of peritonitis, can be life threatening. The assessment of the clinical features of the mice treated with cecal slurry showed an increasing sepsis illness severity after 24 h of observation. The honey-treated mice did not show any difference in the sepsis severity score compared to the nontreated CS-injected group (Figure 3A).

The bacterial burden was evaluated in the blood, peritoneal lavage, and tissues of the CS-treated mice. The bacterial counts in the blood and peritoneal lavage were not different among the CS-injected groups (Figure 3B,C), while they were found to be lower in the spleen (Figure 3D), liver (Figure 3E), and lung (Figure 3F) tissues of the mice pretreated with the different honey samples. Significantly reduced bacterial numbers were found in the liver of septic mice pretreated with the honey samples derived from Evros chestnut and Crete and Epirus fir compared to the CS-treated group (Figure 3E). A significant decrease in the bacterial load was also found in the lungs of the CS-injected mice treated with honey derived from Epirus arbutus, Crete fir, and Manuka honey (Figure 3F).

During sepsis, pathogens are recognized by pattern recognition receptors, leading to the activation of immune cells and the release of pro-inflammatory mediators [31]. The systemic immune response was evaluated by measuring serum cytokine concentrations. Cecal slurry resulted in increased circulating levels of pro-inflammatory cytokines, TNF-α, and IL-6, 4 and 24 h post-injection (Figure 4A–D). None of the honey types tested showed any effect on the production of either TNF-α or IL-6 measured in the serum and peritoneal lavage of the septic mice (Figure 4A–F). 

## 4. Discussion

Endotoxin from Gram-negative bacterial, such as LPS, rapidly induces severe and systemic immune responses that can mimic the initial clinical features of sepsis. Endotoxemia and systemic inflammatory response syndrome are associated with manifestations such as cytokine storm and failure of multiple organs, including the liver [32]. Herein, a model of LPS-induced acute inflammation was used to investigate the potential anti-inflammatory effects of Greek honey types derived from different floral sources and geographical locations. 

Our results showed that all of the Greek honey samples tested possessed anti-inflammatory properties by suppressing the induction of the pro-inflammatory mediators TNFα and iNOS in response to the LPS stimulation. TNF-α and IL-6 reduce the biotransformation capacity of the liver, leading to impairment of its function [32]. Liver function was evaluated by measuring the expression of CYP enzymes that detoxify and eliminate toxic endogenous substances as a result of inflammation. The LPS significantly reduced the expression of CYP enzymes. A lower CYP activity results in a decreased metabolism and the secretion of substrates, leading to the accumulation of these molecules and toxic effects [32]. The administration of fir and chestnut honey the increased expression of CYP enzymes in the liver of the LPS-treated mice, suggesting a potential antioxidant effect of these honey varieties.

The results are consistent with previous studies that examined the anti-inflammatory and immunomodulatory properties of honey. Gelam honey consumption resulted in reduced production of pro-inflammatory mediators, TNF-α, IL-6, NO, COX, and PGE2 due to the inhibition of IκBa degradation and the subsequent inability of NF-κB to translocate into the nucleus in an in vivo model of skin inflammation [33,34]. Gelam honey also improved survival and inhibited the production of pro-inflammatory cytokines (TNF-a and IL-1β) and oxidative stress in an in vivo model of LPS-induced endotoxemia in rats [35]. Manuka honey treatment suppressed LPS-induced inflammation in RAW264.7 macrophages via modulation of p38 MAPK and Erk1/2 [36]. It also inhibited LPS-stimulated pro-inflammatory mediators, such as TNF-α, IL-1β, IL-6, and iNOS [37]. Stingless bee honey administration in rats protected against LPS-induced chronic subclinical systemic inflammation and oxidative stress via the downregulation of inflammatory NF-κB and p38 MAPK signaling and the upregulation of the redox-sensitive transcription factor Nrf2, which enhance the expression of detoxifying and antioxidant enzymes attenuating lipid peroxidation and DNA damage [38]. In accordance, Italian chestnut and eucalyptus honeys suppressed the expression of factors implicated in inflammation and antioxidant defense, including NF-κB, TNF-α, IL-10, iNOS, and Nrf2, in LPS-activated RAW264.7 macrophages [39].

A plethora of in vitro studies on anti-inflammatory properties of honey flavonoids suggest that these compounds are important modulators of inflammatory processes. In vitro evaluation of the total phenolic and flavonoid content as well as the free radical scavenging activities of arbutus, fir, and chestnut honeys revealed potent antioxidant and anti-inflammatory properties [26]. Flavonoids exert their anti-inflammatory action via two mechanisms [9]. Firstly, they possess increased antioxidant capacity linked to direct free radical scavenging [9]. Secondly, flavonoids may act via inhibition of pro-inflammatory enzyme activity, such as iNOS and COX-2 [9]. In addition, they can modulate NF-κB signaling and reduce the expression of pro-inflammatory mediators, namely, TNF-α, IL-1β, IL-6, IL-8, and MCP-1 [9]. For example, luteolin reduced the levels of IL-6, TNF-α, IL-1β, and iNOS in RAW264.7 macrophages activated by LPS and IFN-γ [40]. The incubation of LPS-stimulated human PBMCs with apigenin, luteolin, or chrysin decreased the production of TNF-α, IL-6, and IL-1β [41]. Kaempferol treatment resulted in NF-κΒ suppression and reduced the production of pro-inflammatory mediators il-1β, NO, and PGE2 in LPS-stimulated RAW264.7 macrophages [42].

Complicated intra-abdominal infections such as diffuse peritonitis can occur after diverticulitis or after surgery [43]. These infections are typically polymicrobial, with both aerobic and anaerobic strains [43]. Enterobacterales are most commonly found among Gram-negative bacteria, while Gram-positive Enterococci are particularly prevalent in critically ill patients [43].

Herein, a model of cecal-slurry-induced peritonitis was used to evaluate the antimicrobial activity of Greek arbutus, chestnut, and fir honey. Previous evidence showed an increased H_2_O_2_ content in fir honey compared to arbutus, chestnut, and manuka honey [26]. H_2_O_2_ is a potent antibacterial factor derived from the glucose oxidase-mediated conversion of glucose to gluconic acid under aerobic conditions in diluted honey [10]. Although the H_2_O_2_ concentrations found in all honey types tested (except for fir) were below the minimum concentration required for bacterial cell death [44], the in vitro results showed an increased antibacterial capacity of all the honey types, suggesting the contribution of other bioactive compounds in the honey [26]. Polyphenols can exhibit pro-oxidative activity in the presence of metal ions (i.e., Cu or Fe) and H_2_O_2_, leading to the production of potent hydroxyl radicals (via Fenton reaction), which are responsible for oxidative strand breakage in DNA [45]. The total phenolic content was found to be significantly high in all honey types tested, implying a synergistic antimicrobial effect [26]. Catalase addition revealed that H_2_O_2_ concentration is a key factor in bacterial killing, although other components seem to be involved in the bactericidal ability of honey [26].

The phenolic content of honey consists mainly of phenolic acids and flavonoids that display many different mechanisms to exert their antibacterial activity. For example, caffeic acid eliminates bacterial growth through oxidative stress induction [46], gallic acid leads to bacterial membrane disruption and increased pore formation with consequent leakage of essential intracellular components [47], and p-coumaric acid also disrupts membrane and binds to bacterial genomic DNA to inhibit cellular functions [48]. In addition, common flavonoids, such as quercetin, apigenin, luteolin, and galangin, have the capacity to kill or inhibit bacterial growth through membrane disruption and the inhibition of various processes, including nucleic acid synthesis via helicase or gyrase inhibition, quorum sensing, envelope, and peptidoglycan synthesis [49].

Despite the antimicrobial activity shown in vitro for all honey types, no effect was observed against polymicrobial mouse infection, even for Manuka honey, which was used as the positive control. The bacterial load measured in the blood and peritoneal lavage of the CS-infected mice, treated or not with honey, was at similar levels. Different concentrations of honey could be tested, since we referred to a polymicrobial population, and the sensitivity of some species could be altered due to the interaction with other bacterial species. Evidence from in vitro studies suggests the effective inhibition of honey on polymicrobial culture and biofilm formation [19,50]. The administration of honey in an in vivo model of CLP-induced polymicrobial peritonitis in rats resulted in a lower incidence of postoperative intra-abdominal adhesions and reduced oxidative stress [51]. The antimicrobial activity in terms of bacterial load was not measured for cecum bacterial strains comprised mostly from *Escherichia coli*, *Enterobacter aerogenes*, *Proteus mirabilis*, *Proteus vulgaris*, group D *Streptococcus*, *Enterococcus*, *Staphylococcus aureus,* and anaerobic *Clostridium difficile* and *Bacteroides fragilis* [51]. 

The administration of honey reduced the bacterial load in the tissues of septic mice. This could be an indication that honey has an important role in preventing translocation of microorganisms. In accordance with this finding, honey was previously found to reduce bacterial translocation rates from the intestinal lumen to mesenteric lymph nodes or other extraintestinal tissues in an experimental obstructive jaundice model [52,53]. 

Intestinal disfunction and mucosal injury due to the fact of sepsis could lead to translocation of the intestinal bacteria and toxins [1]. Scientific reports showed that honey and flavonoid administration could prevent colonic inflammation in experimental models of chemical-induced colitis and reduce intestinal lesions, suggesting a protective effect on intestinal barrier integrity [54,55,56]. The oral or systemic administration of polyphenols derived from different plant origins protected rodents from microbial sepsis and endotoxemia, exhibiting anti-inflammatory and vasoprotective properties [57]. Polyphenols reduce iNOS expression, as it was also evidenced in our results, leading to the protection against vascular injury that occurs during sepsis. Polyphenol-mediated vasoprotection could reduce the hyperpermeability of the tissue microvascular system and lower the bacterial burden. Honey and especially flavonoids show a neutralizing and inhibitory effect on bacterial virulence factors, such as toxins and other bacterial products [49,58]. Quercetin and myricetin have been found to inhibit bacterial hyaluronidase that could mediate hyaluronal degradation and increased connective tissue permeability that could help bacteria to evade and disseminate in connective tissues [59].

## 5. Conclusions

In conclusion, our results showed that treatment with Greek arbutus, chestnut, and fir honeys resulted in the suppression of inflammatory responses in mice with acute endotoxemia by reducing the production of TNF-α and iNOS. In addition, the administration of Greek honeys decreased the bacterial translocation to tissues during polymicrobial peritonitis.

## Figures and Tables

**Figure 1 microorganisms-10-02374-f001:**
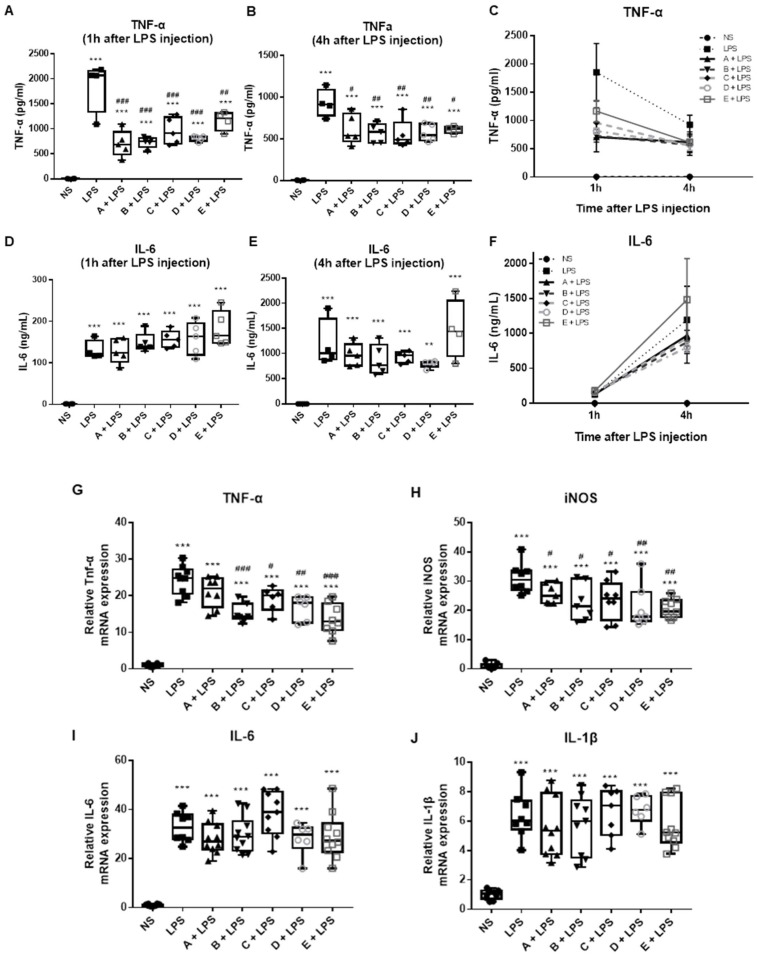
Greek honey types exert anti-inflammatory properties: serum (**A**–**C**) TNF-α and (**D**–**F**) IL-6 levels, measured 1 and 4 h after LPS injection. Liver mRNA expression of (**G**) TNF-α, (**H**) iNOS, (**I**) IL-6, and (**J**) IL-1β. All graphs represent measurements from 5 mice per group; the horizontal line in the box plots represents the median, and the whiskers represent the minimum and maximum. In the post-hoc test, the normal saline-treated group (NS) was selected as the control group for multiple comparisons with the LPS-injected groups: ** *p* < 0.01 and *** *p* < 0.001. The NS- vs. the LPS-injected groups or the LPS group was selected as the control group for multiple comparisons with the honey-treated groups: ^#^
*p* < 0.05, ^##^
*p* < 0.01, and ^###^
*p* < 0.005; the honey-treated group + LPS vs. the LPS control group. The absence of an asterisk or hash indicates no statistical significance. A: Epirus arbutus; B: Evros chestnut; C: Epirus fir; D: Crete fir; E: Manuka honey.

**Figure 2 microorganisms-10-02374-f002:**
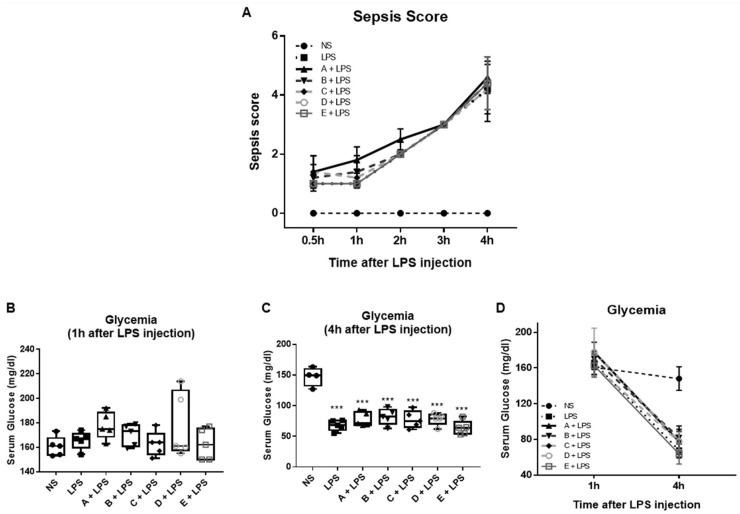
Effect of the Greek honey samples on the severity of LPS-induced endotoxemia: (**A**) severity score of LPS-induced sepsis; (**B**–**D**) serum glucose levels at 1 and 4 h after LPS challenge; liver mRNA expression of (**E**) CYP1A1, (**F**) CYP2B10, (**G**) CYP3A11, and (**H**) CYP3A25. All graphs represent measurements from 5 mice per group; the horizontal line in the box plots represents the median, and the whiskers represent the minimum and maximum. In the post-hoc test, the normal saline-treated group (NS) was selected as the control group for multiple comparisons with the LPS-injected groups: *** *p* < 0.001. The NS- vs. the LPS-injected groups or the LPS group was selected as the control group for multiple comparisons with the honey-treated groups: ^#^
*p* < 0.05, honey-treated group + LPS vs. LPS control group. The absence of an asterisk or hash indicates no statistical significance. A: Epirus arbutus; B: Evros chestnut; C: Epirus fir; D: Crete fir; E: Manuka honey. Clinical scoring system: 0 = no abnormal clinical signs; 1 = ruffled fur but lively; 2 = ruffled fur, moving slowly, hunched, and sick; 3 = ruffled fur, squeezed eyes, hardly moving, down, and very sick; 4 = same as 3 but with incontinence; 5 = moribund.

**Figure 3 microorganisms-10-02374-f003:**
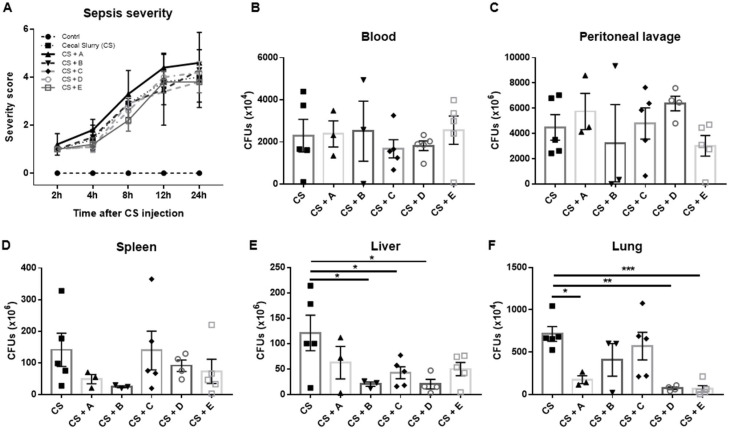
Effect of the Greek honey types’ administration on CS-induced sepsis: (**A**) severity score of sepsis and the bacterial load in the (**B**) blood, (**C**) peritoneal lavage, (**D**) spleen, (**E**) liver, and (**F**) lungs of CS-injected mice. All graphs represent measurements from 5 mice per group and show the mean ± SEM. In the post-hoc test, the CS-injected group was selected as the control group for multiple comparisons with the honey-treated groups: * *p* < 0.05, ** *p* < 0.01, and *** *p* < 0.001. The absence of an asterisk indicates no statistical significance. A: Epirus arbutus; B: Evros chestnut; C: Epirus fir; D: Crete fir; E: Manuka honey. Clinical scoring system: 0 = no abnormal clinical signs; 1 = ruffled fur but lively; 2 = ruffled fur, moving slowly, hunched, and sick; 3 = ruffled fur, squeezed eyes, hardly moving, down, and very sick; 4 = same as 3 but with incontinence; 5 = moribund.

**Figure 4 microorganisms-10-02374-f004:**
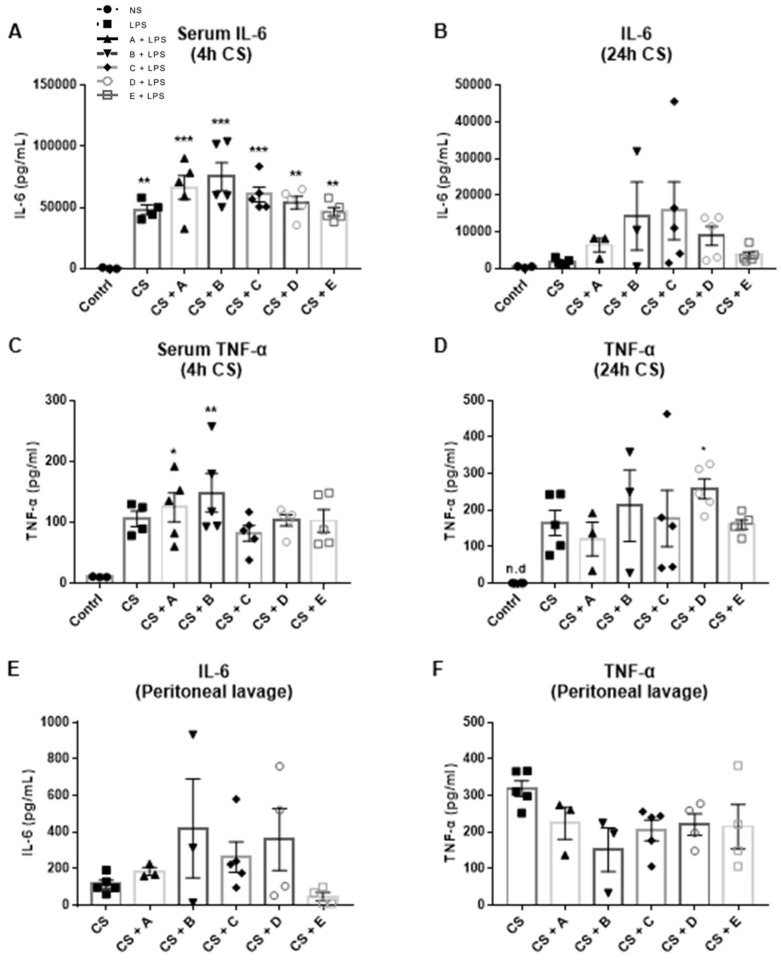
Effect of the Greek honey on CS-induced inflammation: serum (**A**,**B**) IL-6 and (**C**,**D**) TNF-α levels measured 4 and 24 h after cecal slurry injection; (**E**) IL-6 and (**F**) TNF-α levels in the peritoneal lavage of septic mice. All graphs represent measurements from 5 mice per group and show the mean ± SEM. In the post-hoc test, the dextrose-injected group (contrl) was selected as the control group for multiple comparisons with the honey-treated groups: * *p* < 0.05, ** *p* < 0.01, and *** *p* < 0.001, CS-injected vs. control group. The absence of an asterisk indicates no statistical significance. A: Epirus arbutus; B: Evros chestnut; C: Epirus fir; D: Crete fir; E: Manuka honey.

## Data Availability

Not applicable.

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
