# Peer review of "Anti-Inflammatory and Antibacterial Effects and Mode of Action of Greek Arbutus, Chestnut, and Fir Honey in Mouse Models of Inflammation and Sepsis"

_microorganisms, 2022, doi:10.3390/microorganisms10122374_

Round 1

Reviewer 1 Report

Please find the attach file(major revision)

Author Response

Dear Reviewer

We thank you for your comments that help improving our manuscript

Please find hereby responses to your comments.

Sincerely,

Elisavet Stavropoulou

Comment: Abstract: Abstract includes background, objective, materials and methods, results, and a brief conclusion.

Response: We have now restructured the abstract according the Reviewer’s suggestion. In abstract additional background was added (lines 22-26) and a brief conclusion was included (lines 46-49)

Comment: Introduction:

  • Line 59 and 60: which vary according to the botanical origin, honey bee species, environmental conditions and geographical origins [4-6].
    • Taha, E-K.A., Al-Kahtani, S.N.; Taha, R. (2018). Comparison of pollen spectra and amount of mineral content in honey produced by Apis florea F. and Apis mellifera L. Journal of the Kansas Entomological Society, 91(1): 51-58.
    • Taha, E-K.A., Al-Kahtani, S.N.; Taha, R. (2021). Comparison of the physicochemical characteristics of sidr (Ziziphus spp.) honey produced by Apis florea F. and Apis mellifera L. Journal of Apicultural Research, 60(3): 470–477.
  • Line 97 and 98: Please italic the scientific names.

Response: We have now included the information requested on honey bee species, along with the suggested references (lines 68-69). In addition, all scientific names are now are italics.

Comments: Materials and Methods:

  • Line 110: replace 4-h period to be: 4 hrs.
  • Line 112: 1 and 4 hrs.
  • Line 113: -80 °C
  • Line 121: replace for a 24-h period to be: for 24 hrs.
  • Line 138: Statistical analysis: What is the type of experimental design?
  • Line 139: The results were analyzed using two- or one-way ANOVA?????? The authors have used only one factor, so the one-way ANOVA should be follow.

Response: We thank the reviewer for the suggestions. We have proceeded with all suggested changes in the revised manuscript, which are highlighted in yellow.

Regarding the comment on the statistics (lines 138 and 139) we have now described in more detail the statistical analysis used (lines 147-153). Statistical analysis was performed using one way ANOVA or two way ANOVA according to the type of data compared, which is now described in material and methods and in the revised legends of the figures.

Comment: Results:

  • In Figs., letters above the bars should be added.
  • The standard error in the 3rd column in Fig. 3C seems more than the mean value, I think something wrong.
  • The standard error in the 3rd column in Fig. 4B and E seems near the mean value, I think something wrong.
  • The high standard error influences the significance among treatments.

Response: Since we present the data showing each individual sample and we have performed statistics comparing samples with the respective control, we opted to present the comparisons highlighting the significant differences using stars rather than having letters above each bar, which would make the figure more difficult for the reader to understand. To allow the reader understand the comparisons, more information is now included in the Figure legends, where we explicitly indicated that absence of asterisk or hash indicates no statistical significance.

The 3rd column in figure 3C and 3rd column in figure 4B and E represent mean ± standard error of the mean that is automatically depicted by the software (GraphPad Prizm) through data analysis.

Regarding the high standard error, it is indeed an innate limitation of in vivo experiments but in the cases that the effect was profound this was very obvious and standard error was smaller. In the cases that the standard error was high we did not expect to find significant differences since the other parameters tested also showed no difference.

Comment: Discussion:

Discussion needs more improvements.

Response: We thank the reviewer for the suggestion. Discussion has been expanded providing background mechanism information with related references (Lines 291-305; lines 312-335 and lines 355-369.

Reviewer 2 Report

A professional editing is required to improve the manuscript. If possible, the Discussion can be expanded a bit more by adding more background mechanism with references.   

Author Response

Dear Reviewer

We thank you for your comments that help improving our manuscript

Please find hereby responses to your comments.

Sincerely,

Elisavet Stavropoulou

Comment: A professional editing is required to improve the manuscript. If possible, the Discussion can be expanded a bit more by adding more background mechanism with references.  

Response: Discussion was expanded by including additional information on background mechanism with references, as suggested. (Lines 291-305; lines 312-335 and lines 355-369). In addition, the document was professionally edited.

Round 2

Reviewer 1 Report

Thank you for modifying the manuscript.

The references need to modify according to the Journal format.

Author Response

The references were modified as well following the journal style